# Overcoming the nutritional immunity by engineering iron-scavenging bacteria for cancer therapy

**Sin-Wei Huang[1], See-Khai Lim[1], Yao-An Yu[1,2], Yi-Chung Pan[1], Wan-Ju Lien[1], Chung-Yuan Mou[3]\*, Che-Ming Jack Hu[1,2]\*, Kurt Yun Mou[1†]**

[1]Institute of Biomedical Sciences, Academia Sinica, Taipei, Taiwan; [2]Doctoral Degree Program of Translational Medicine, National Yang Ming Chiao Tung University and Academia Sinica, Taipei, Taiwan; [3]Department of Chemistry, National Taiwan University, Taipei, Taiwan

**\*For correspondence:**
cymou@ntu.edu.tw (C-YM);
chu@ibms.sinica.edu.tw (C-MJH)

[†]Deceased

**Competing interest:** The authors declare that no competing interests exist.

**Abstract** Certain bacteria demonstrate the ability to target and colonize the tumor microenvironment, a characteristic that positions them as innovative carriers for delivering various therapeutic agents in cancer therapy. Nevertheless, our understanding of how bacteria adapt their physiological condition to the tumor microenvironment remains elusive. In this work, we employed liquid chromatography-tandem mass spectrometry to examine the proteome of *E. coli* colonized in murine tumors. Compared to *E. coli* cultivated in the rich medium, we found that *E. coli* colonized in tumors notably upregulated the processes related to ferric ions, including the enterobactin biosynthesis and iron homeostasis. This finding indicated that the tumor is an iron-deficient environment to *E. coli*. We also found that the colonization of *E. coli* in the tumor led to an increased expression of lipocalin 2 (LCN2), a host protein that can sequester the enterobactin. We therefore engineered *E. coli* in order to evade the nutritional immunity provided by LCN2. By introducing the IroA cluster, the *E. coli* synthesizes the glycosylated enterobactin, which creates steric hindrance to avoid the LCN2 sequestration. The IroA-*E. coli* showed enhanced resistance to LCN2 and significantly improved the anti-tumor activity in mice. Moreover, the mice cured by the IroA-*E. coli* treatment became resistant to the tumor re-challenge, indicating the establishment of immunological memory. Overall, our study underscores the crucial role of bacteria's ability to acquire ferric ions within the tumor microenvironment for effective cancer therapy.

## eLife assessment

This **valuable** study combines proteomics and a mouse model to reveal the importance of iron uptake in bacterial therapy for cancer. The evidence presented is **convincing**. Notably, the authors showed upregulation of iron uptake of bacteria significantly inhibits tumor growth in vivo. This paper will be of interest to a broad audience including researchers in cancer biology, cell biology, and microbiology.

## Introduction

Bacterial therapy has re-emerged as a promising modality for cancer treatment, building on the pioneering works of William Coley in the late 19[th] century (*Carlson et al., 2020*). As a living therapeutic agent, bacteria offer several advantages over traditional cancer treatments, including (1) active tumor targeting (*Yamamoto et al., 2016*; *Kim et al., 2023*), (2) tumor colonization (*Westphal et al., 2008*; *Weibel et al., 2008*), (3) immune system stimulation (*Qiu et al., 2020*; *Yang et al., 2023*), and

(4) great engineerability for customized functionalities (*Gurbatri et al., 2020*; *Hu et al., 2022*; *Canale et al., 2021*). The tumor microenvironment (TME) provides unique cues, such as hypoxia, low pH values, and immune suppressors, which facilitate the selective targeting and colonization of certain bacteria, including *E. coli* (*Ryan et al., 2009*; *Flentie et al., 2012*). However, the restricted resource in the TME may pose challenges for bacterial survival and growth, potentially limiting their anti-tumor efficacy. While the molecular physiology of *E. coli* has been extensively studied under well-controlled laboratory conditions (*Han and Lee, 2006*; *Mateus et al., 2020*), it is unclear how *E. coli* adapts to the nutrition-limited and immune-responsive environment in tumors. An in-depth understanding of the intricate relationship between the TME and the adaptation of colonized bacteria may provide hints to unlock the full potential of bacterial therapy against cancers.

It is well-established that the innate and adaptive immunity are sequentially activated upon bacterial infection in humans. However, even before the innate immune response commences, a critical defense mechanism known as 'nutritional immunity' serves as the first line of protection to hinder the bacterial infection. Nutritional immunity is employed by the host organism through restricting the availability of essential nutrients to the invading pathogens (*Weinberg, 1975*; *Murdoch and Skaar, 2022*). For example, humans have evolved specialized proteins to chelate trace minerals, such as iron, zinc, and manganese, keeping their free-form concentrations at very low levels within the body (*Andrews and Schmidt, 2007*; *Kambe et al., 2015*; *Roth et al., 2013*). In response, pathogens have counter-evolved mechanisms to evade nutritional immunity. For instance, a variety of siderophores are developed by bacteria to acquire ferric ions from the host (*Kramer et al., 2020*). Intriguingly, humans have further adapted by evolving siderophore-sequestering proteins, such as LCN2, for bacterial inhibition (*Singh et al., 2015*; *Bachman et al., 2009*). These co-evolutionary events highlight the importance of nutritional immunity in combating bacteria (*Golonka et al., 2019*). Previous studies on bacterial cancer therapy have rarely examined the role of nutritional immunity, which may limit bacteria's therapeutic efficacy.

In this work, we aimed to understand how bacteria modulate their physiological states at the molecular level in response to the tumor microenvironment. We employed liquid chromatography-tandem mass spectrometry (LC-MS/MS) for a quantitative comparison of the proteome between *E. coli* cultured in a nutrient-rich medium and *E. coli* colonized in tumors. We found that *E. coli* colonized in tumors dramatically increases the protein expressions involved in the enterobactin biosynthesis and the iron ion homeostasis. This finding suggested that *E. coli* was stressed in an iron-deficient environment. Driven by this discovery, we hypothesized that enhancing the iron acquisition ability of *E. coli* might improve its anti-tumor activity. We engineered *E. coli* with enhanced iron-scavenging capacity by introducing LCN2 blockers, including cyclic-di-GMP and glycosylated enterobactin. The engineered bacteria evaded the nutritional immunity and achieved complete tumor emission in mouse models, highlighting strategy for improving anti-tumor bacterial therapy through circumvention of nutritional immunity.

## Results

### Tumor is an iron-deficient microenvironment for bacterial colonization

While many bacteria are known to colonize and proliferate in the TME, it remains elusive how bacteria adapt to such a nutrition-limited environment as compared to a nutrition-rich one. To this end, we performed quantitative LC-MS/MS experiments to compare the proteome of *E. coli* colonized in the murine tumors or cultured in the rich medium (LB broth) (*Figure 1—figure supplement 1*). Not surprisingly, there are many more proteins enriched in the rich medium than in the tumor condition. The Venn diagram and the volcano blot revealed hundreds of protein IDs enriched in the rich medium condition (*Figure 1a and b*). The Gene Ontology (GO)-term analysis revealed that many of these proteins are associated with the machineries of biosynthetic processes, cell division, and energy production (*Figure 1c*), reflecting that the *E. coli* was under a highly proliferative state in the rich medium. Interestingly, there were also 71 proteins that were preferentially expressed in the tumor condition over the rich medium condition. The GO-term analysis and the hierarchical clustering revealed that many of these proteins are involved in the processes of enterobactin synthesis and iron ion homeostasis (*Figure 1d and e*, and *Figure 1—figure supplement 2*). *Figure 1f* showed that the individual proteins in these two processes were markedly up-regulated in the tumor condition. These proteins are known

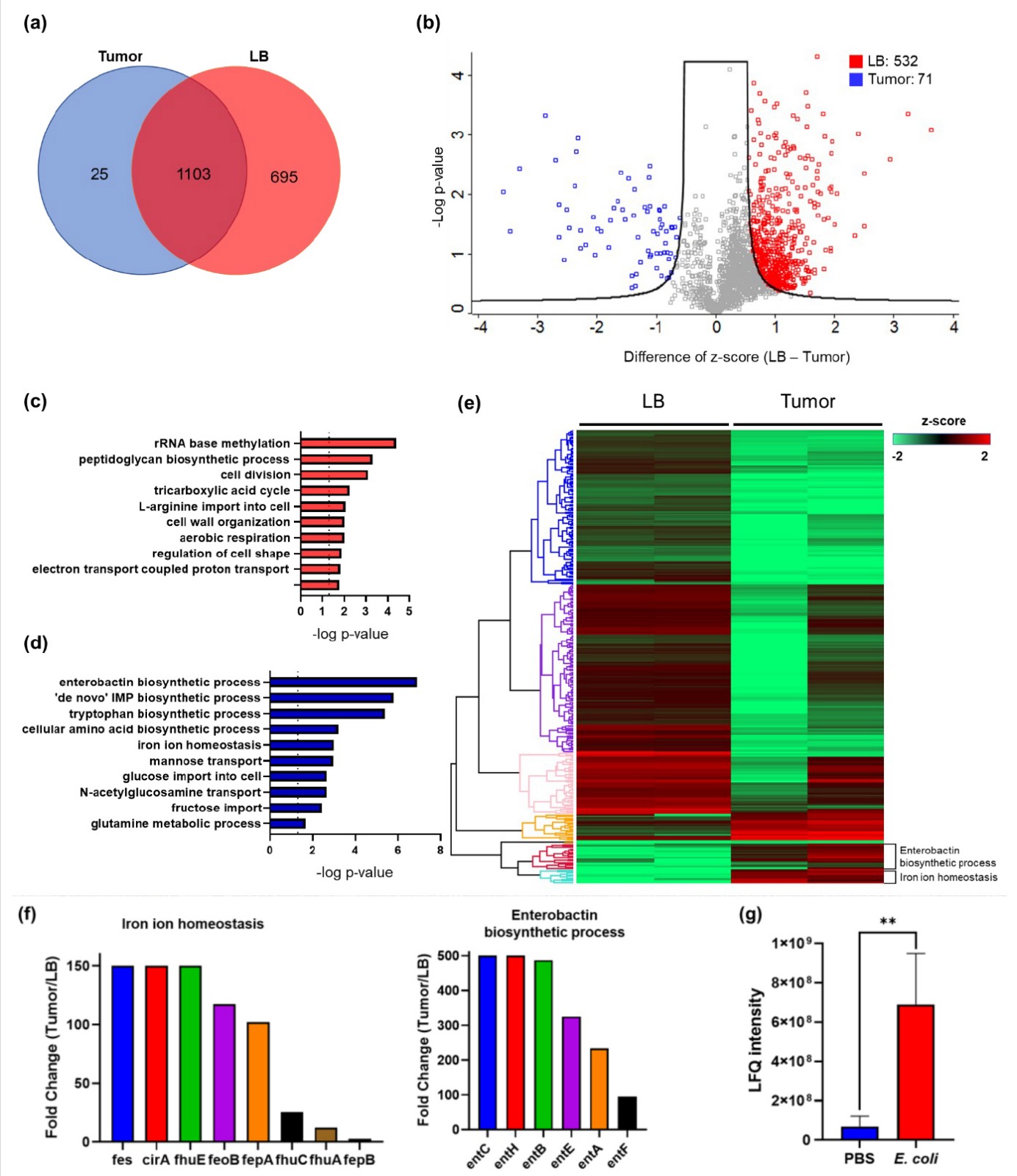

**Figure 1.** The quantitative proteomic analysis of *E. coli* in the rich medium and the tumor microenvironment. (**a**) The Venn diagram of the *E. coli* protein IDs identified in the rich medium and in the tumor microenvironment (TME). (**b**) The volcano plot of the *E. coli* protein IDs quantified in the rich medium and in the TME. (**c**) The Gene Ontology (GO)-term analysis of the protein IDs enriched in the rich medium condition. (**d**) The GO-term analysis of the protein IDs enriched in the tumor condition. (**e**) The hierarchical clustering analysis of the protein IDs identified in the rich medium and in

*Figure 1 continued*

the TME. Each column is a biological replicate. (**f**) Left: the fold changes of individual proteins in the iron ion homeostasis process. These proteins are involved in transporting or processing the iron ions. Right: the fold changes of individual proteins in the enterobactin biosynthesis process. (**g**) Label-free quantification of lipocalin 2 (LCN2) in the tumors with and without *E. coli* inoculation. The error bars represent mean ± SD. Statistical analyses were performed by Student's t-test (**p<0.01).

The online version of this article includes the following figure supplement(s) for figure 1:

**Figure supplement 1.** Workflow of quantitative proteomics for *E. coli* cultured in rich medium and *E. coli* colonized in murine tumors.

**Figure supplement 2.** STRING network analysis.

to be tightly controlled by the iron sensing system in *E. coli*, and only become up-regulated under the stress of iron deficiency (*Seo et al., 2014*). It is worth mentioning that while enterobactin facilitates the uptake of ferric ions into bacteria, the host immune cells can counteract by secreting a protein called LCN2, which possesses a specialized pocket to bind and sequester enterobactin (*Fischbach et al., 2006*). To investigate this possibility, we employed LC-MS/MS and analyzed the LCN2 expression in the tumors with or without *E. coli* colonization. Indeed, we observed a significant up-regulation of LCN2 expression in the tumors with *E. coli* inoculation (*Figure 1g*). Collectively, our data suggest that the tumor is an iron-deficient environment for *E. coli* colonization.

## Cyclic-di-GMP-producing *E. coli* synergizes with iron chelators for cancer therapy

Based on the proteomic findings, we aimed to focus our therapeutic strategies on modulating the iron competition between bacteria and cancer cells in the tumor. First, we hypothesize that an iron chelator that lowers the effective pool concentration of iron may provide a selection pressure, which disfavors the growth of cancer cells over bacteria. We tested three iron chelators, deferoxamine, ciclopirox, and VLX600, which have been approved for clinical trials or medical applications (*Lang et al., 2019*; *Qi et al., 2020*; *Fryknäs et al., 2016*). All three chelators showed high cytotoxicity toward the cancer cells, suggesting the essential role of irons for the survival of mammalian cells (*Figure 2a*). On the other hand, *E. coli* can better tolerate these iron chelators at relatively high concentrations. Among them, VLX600 was the most potent drug against the cancer cells (IC50=0.33 µM) and provided the largest therapeutic window (280-fold difference) between the cancer cells and *E. coli*. In addition to the iron chelator, we also attempted to find approaches for counteracting the enterobactin seques-tering function of LCN2. It has been reported that cyclic di-GMP (CDG) can block the binding between LCN2 and enterobactin, therefore, restoring the functionality of enterobactin (*Li et al., 2015*). In our bacterial culture study, we validated that the addition of CDG can enhance bacteria survival in the presence of LCN2 (*Figure 2—figure supplement 1*), thus prompting our effort to prepare CDG-expressing bacteria. We engineered the *E. coli* by introducing a plasmid that carries the gene of diguanylate cyclase (DGC), an enzyme responsible for catalyzing the biosynthesis of cyclic di-GMP (CDG) (*Lv et al., 2019*). We showed that the *E. coli* transformed with the DGC plasmid (hereafter referred to DGC-*E. coli*) actively synthesized CDG and secreted it into the supernatant as detected by the LC/MS-MS mass spectrometry (*Figure 2b*). It is worth noting that CDG is also a potent ligand for the STING pathway, which can stimulate anti-tumor immunity (*Diner et al., 2013*; *Krasteva and Sondermann, 2017*; *Chattopadhyay et al., 2020*). When applying the DGC-*E. coli* supernatant to the macrophages, the macrophages enhanced the IFN-β secretion, indicating the activation of the STING pathway by CDG (*Figure 2c*).

We evaluated the anti-tumor activity of VLX600 and DGC-*E. coli* in a syngeneic mouse model. The MC38 tumor-bearing mice received VLX600 and/or DGC-*E. coli* as depicted in *Figure 2d*. The VLX600 monotherapy only marginally suppressed the tumor growth as compared to the PBS control (*Figure 2e*). The DGC-*E. coli* monotherapy, although inhibited the tumor progression to a certain degree, did not show superior efficacy than the wild-type *E. coli* without the DGC plasmid transforma-tion. Strikingly, the combination of DGC-*E. coli* and VLX600 showed significantly improved efficacy as compared to the individual mono-therapies. The tumor sizes were greatly suppressed, and 2 out of 4 mice achieved complete remission (*Figure 2e and f*). Of note, the combination of the wild-type *E. coli* and VLX600 did not result in such an improved activity, indicating that the CDG expression contrib-uted to bacteria's synergism with the iron-chelating VLX600. The combination of CDG-expressing

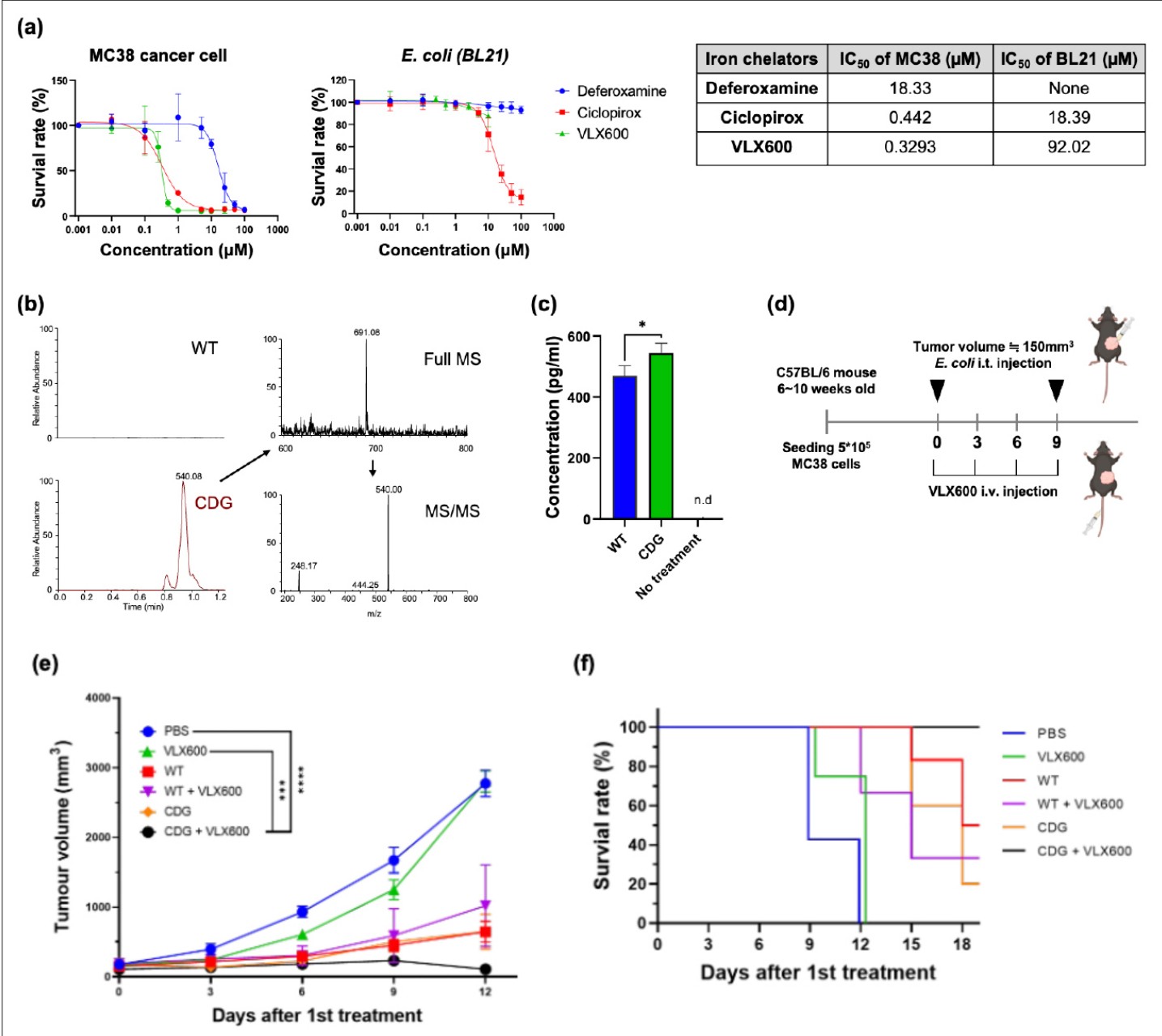

**Figure 2.** Combination of iron chelator and diguanylate cyclase (DGC)-*E. coli* for cancer therapy. (**a**) Toxicity profiles of various iron chelators against MC38 cancer cells and *E. coli*. (**b**) Identification of cyclic-di-GMP secretion from DGC-*E. coli* by liquid chromatography-tandem mass spectrometry (LC-MS/MS). The precursor ion and the fragmented product ions correspond to the correct molecule weights of cyclic di-GMP (CDG). (**c**) IFN-β secretion by RAW264.7 cells treated with the supernatants from wild-type *E. coli* or DGC-*E. coli*. (**d**) Schematic illustration of mouse treatments. The DGC-*E. coli* was intratumorally delivered on Day 0 and Day 9, whereas VLX600 was intravenously administrated every three days from Day 0 to Day 9. (**e**) Tumor growth curve for various treatment groups. The complete remission was only achieved in the CDG + VLX600 group (CR = 2/4). (**f**) The Kaplan-Meier analysis for different treatment groups. The mouse was considered dead when the tumor volume exceeded 1500 mm³. The error bars represent mean ± SD. Statistical analyses were performed by Student's t-test (*p<0.05).

The online version of this article includes the following figure supplement(s) for figure 2:

**Figure supplement 1.** High concentration of cyclic-di-GMP partially blocks lipocalin 2 (LCN2).

**Figure supplement 2.** Tumor re-challenge experiments in mice.

bacteria and VLX600 also established robust anticancer adaptive immunity as the mice cured by the combinatorial therapy showed no tumor development in a rechallenge study (*Figure 2—figure supplement 2*). These results highlight the benefit of enhancing anticancer bacterial therapy through nutritional immunity manipulation.

## Salmochelin-secreting *E. coli* significantly impeded tumor growth

Encouraged by the results shown above, we sought to engineer *E. coli* with a more specific iron-scavenging functionality to assess its benefit against nutritional immunity. In nature, bacteria have evolved a strategy to block LCN2 through the expression of glycosylated enterobactin or salmochelin (*Fischbach et al., 2006*). Some pathogenic *E. coli* strains carry a gene cluster called IroA, which consists of five genes to perform enterobactin glycosylation and processing. The sugars on the enterobactin create steric hindrance to the LCN2 pocket, thereby abolishing LCN2 binding. To investigate this effect, we cloned the IroA cluster into a plasmid and transformed it into a non-IroA-carrying *E. coli* strain BL21(DE3). A non-enterobactin-expressing △entE-*E. coli* strain, which is particularly susceptible to LCN2 binding, was employed for comparison. We incubated the *E. coli* with varying concentrations of LCN2 and measured their viability by colony formation assay. *Figure 3a* shows that the *E. coli* without the IroA plasmid transformation (referred as WT-*E. coli*) was sensitive to LCN2, whereas the *E. coli* transformed with the IroA plasmid (referred as IroA-*E. coli*) was significantly more resistant to LCN2. The IroA-*E. coli* also showed stronger potency in acquisition of the iron ions than the WT-*E. coli* while the LCN2 was presented in the environment (*Figure 3b*). In line with this observation, we found that the enterobactin (including the glycosylated form) extracted from the IroA-*E. coli* was more cytotoxic to the cancer cells than that extracted from the WT *E. coli* (*Figure 3c*).

In mouse models of colon, breast, and melanoma cancers, the IroA-*E. coli* was significantly more efficacious than the WT-*E. coli*. In a MC38 mouse colon cancer model, 6 out of 10 mice treated by IroA-*E. coli* achieved complete remission, whereas none of the mice treated by the WT-*E. coli* experienced a cure (*Figure 3d–f*, and *Figure 3—figure supplement 1a*). In two other mouse models of E0771 breast cancer and B16F10 melanoma, the IroA-*E. coli* demonstrated improved anti-tumor ability as compared to WT bacteria (*Figure 3g–j*, and *Figure 3—figure supplement 1b–c*). It should be pointed out LCN2 expression has been shown to elevate the aggressiveness of breast (*Yang et al., 2009*), melanoma (*Adler et al., 2023*), and colon cancers (*Chaudhary et al., 2021*), suggesting that WT bacteria may show reduced anticancer activity in more aggressive cancer types due to higher iron competition. Overall, we showed that the IroA cluster equips *E. coli* with an effective iron-scavenging capability, exerting a potent anti-tumor effect in the LCN2-rich tumor microenvironment.

To evaluate the safety of IroA-E. coli in comparison to WT bacteria, we intravenously administered the bacteria and conducted serial whole blood analyses on days 1, 3, and 7 post-injection to assess bacterial burden and blood cell counts (*Figure 3k–l*). For both WT-*E. coli* and IroA-*E. coli*, bacterial burden was undetectable in the blood. Additionally, whole blood cell analysis indicated that IroA-*E. coli* did not adversely affect the immune system within the circulatory system compared to WT-*E. coli*. By day 7, all treatment groups had comparable blood cell counts to the untreated (UT) groups. These findings demonstrate that IroA-*E. coli* could enhance anti-tumor treatment without incurring additional risks of bacteremia or sepsis relative to WT-*E. coli*.

## IroA-*E. coli* is less iron-deficient than WT-*E. coli* in the tumor microenvironment

Because the salmochelin secreted by IroA-*E. coli* is resistant to the sequestration of LCN2, we speculated that IroA-*E. coli* could have ameliorated the iron deficiency problem in the TME. To verify this speculation, we quantitatively compared the proteome of WT-*E. coli* and IroA-*E. coli* colonized in the tumors (*Figure 4a*). The GO-term analysis revealed that the many proteins enriched in WT-*E. coli* belong to the enterobactin biosynthesis process and the iron homeostasis. *Figure 4b* and *Figure 4c* showed the fold changes of the individual proteins in these two terms. Strikingly, all of these proteins were expressed at a much higher level in WT-*E. coli* than in IroA-*E. coli*. These results suggest that IroA-*E. coli* was less stressed by the iron-deficient environment in the TME, which is in accordance with IroA-*E.coli*'s superior iron-scavenging ability from the glycosylated enterobactin. Besides the proteome of *E. coli*, we also examined the proteome changes in the host cells. We found that two iron-related proteins in mice, transferrin and transferring receptor, were elevated in the IroA-*E. coli*

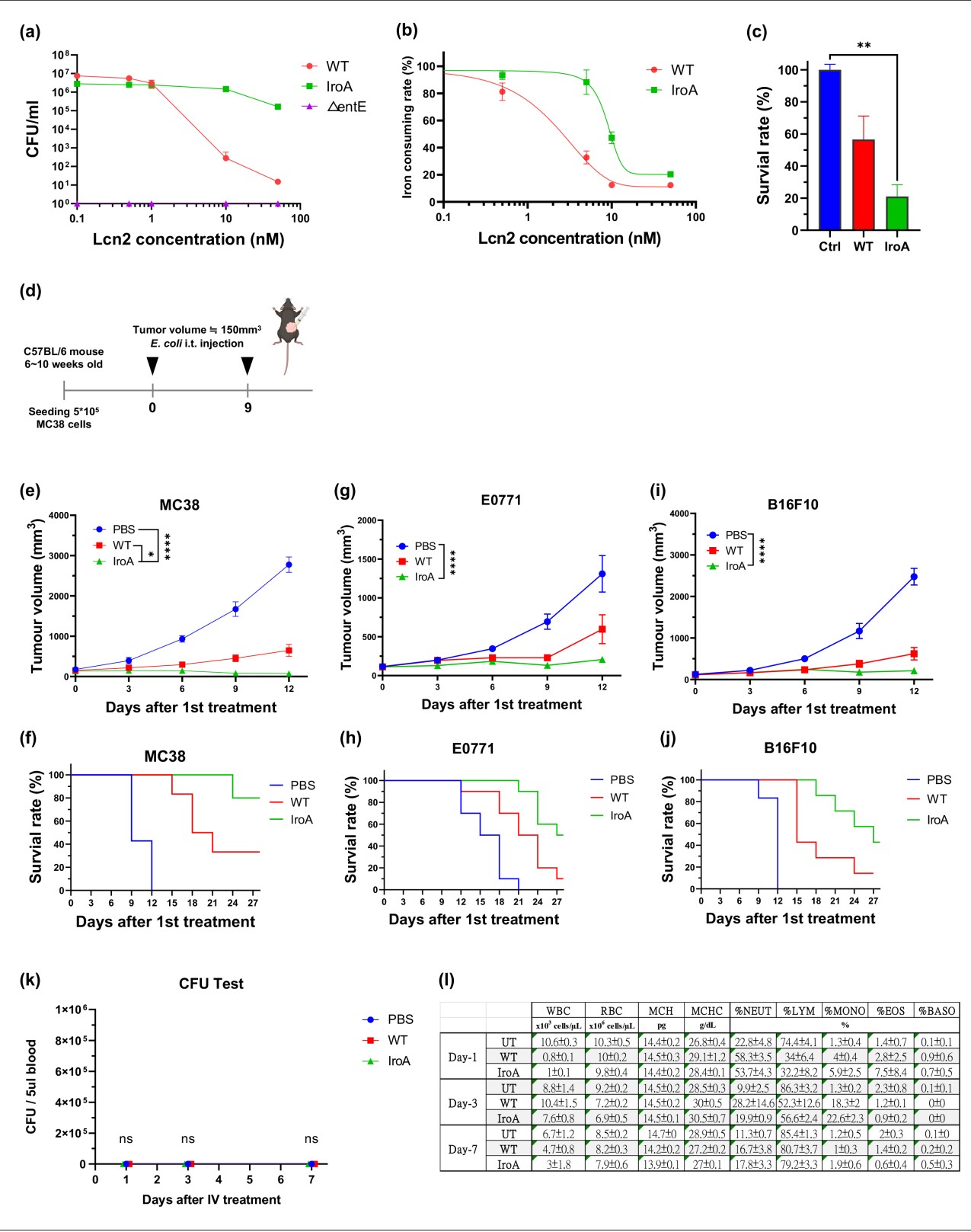

Table (l):

| | | WBC | RBC | MCH | MCHC | %NEUT | %LYM | %MONO | %EOS | %BASO |
|---|---|---|---|---|---|---|---|---|---|---|
| | | x10³ cells/µL | x10⁶ cells/µL | pg | g/dL | % | | | | |
| Day-1 | UT | 10.6±0.3 | 10.3±0.5 | 14.4±0.2 | 26.8±0.4 | 22.8±4.8 | 74.4±4.1 | 1.3±0.4 | 1.4±0.7 | 0.1±0.1 |
| | WT | 0.8±0.1 | 10±0.2 | 14.5±0.3 | 29.1±1.2 | 58.3±3.5 | 34±6.4 | 4±0.4 | 2.8±2.5 | 0.9±0.6 |
| | IroA | 1±0.1 | 9.8±0.4 | 14.4±0.2 | 28.4±0.1 | 53.7±4.3 | 32.2±8.2 | 5.9±2.2 | 7.5±8.4 | 0.7±0.5 |
| Day-3 | UT | 8.8±1.4 | 9.2±0.2 | 14.5±0.2 | 28.5±0.3 | 9.9±2.5 | 86.3±3.2 | 1.3±0.2 | 2.3±0.8 | 0.1±0.1 |
| | WT | 10.4±1.5 | 7.2±0.2 | 14.5±0.2 | 30±0.5 | 28.2±14.6 | 52.3±12.6 | 18.3±2 | 1.2±0.1 | 0±0 |
| | IroA | 7.6±0.8 | 6.9±0.5 | 14.5±0.1 | 30.5±0.7 | 19.9±0.9 | 56.6±2.4 | 22.6±2.3 | 0.9±0.2 | 0±0 |
| Day-7 | UT | 6.7±1.2 | 8.5±0.2 | 14.7±0 | 28.9±0.5 | 11.3±0.7 | 85.4±1.3 | 1.2±0.5 | 2±0.3 | 0.1±0 |
| | WT | 4.7±0.8 | 8.2±0.3 | 14.2±0.2 | 27.2±0.2 | 16.7±3.8 | 80.7±3.7 | 1±0.3 | 1.4±0.2 | 0.2±0.2 |
| | IroA | 3±1.8 | 7.9±0.6 | 13.9±0.1 | 27±0.1 | 17.8±3.3 | 79.2±3.3 | 1.9±0.6 | 0.6±0.4 | 0.5±0.3 |

**Figure 3.** Characterization of IroA-*E. coli* for anti-tumor activity. (**a**) *E. coli* viability in varying concentrations of lipocalin 2 (LCN2) protein. The ΔentE strain, which could not generate enterobactin, was used as a negative control. (**b**) Iron-consuming ability of *E. coli* determined by the chrome azurol S (CAS) assay reagent. (**c**) Cytotoxicity of enterobactin on the MC38 colon cancer cells. The enterobactin was extracted from an equal supernatant volume of the wild-type (WT)-*E. coli* or the IroA-*E. coli* culture. The extraction buffer (DMSO) was used as a negative control. (**d**) Treatment schedule of IroA-*E.*

*Figure 3 continued on next page*

*Figure 3 continued*

*coli* in tumor-bearing mice. Two intratumoral injections were administered on Day 0 and Day 9. (**e**) Tumor growth curves across various treatment groups. (**f**) The Kaplan-Meier analysis for the mice in different treatment groups. (**g**) E0771 breast tumor growth curves for the different treatment groups. (**h**) Survival curves for mice in different treatment groups. (**i**) B16F10 melanoma tumor growth curves for the different treatment treatment groups. (**j**) Survival curves for mice in the different treatment groups. (**k**) The bacterial burden from the blood of mice on days 1, 3, and 7 following intravenous administration with different bacteria. (**l**) Whole blood cell analyses for the different treatment groups. The error bars represent mean ± SD. Statistical analyses were performed by one-way ANOVA (\*\*p<0.01).

The online version of this article includes the following figure supplement(s) for figure 3:

**Figure supplement 1.** The tumor growth curves of individual mice in MC38, E0771, B16F10 tumor models.

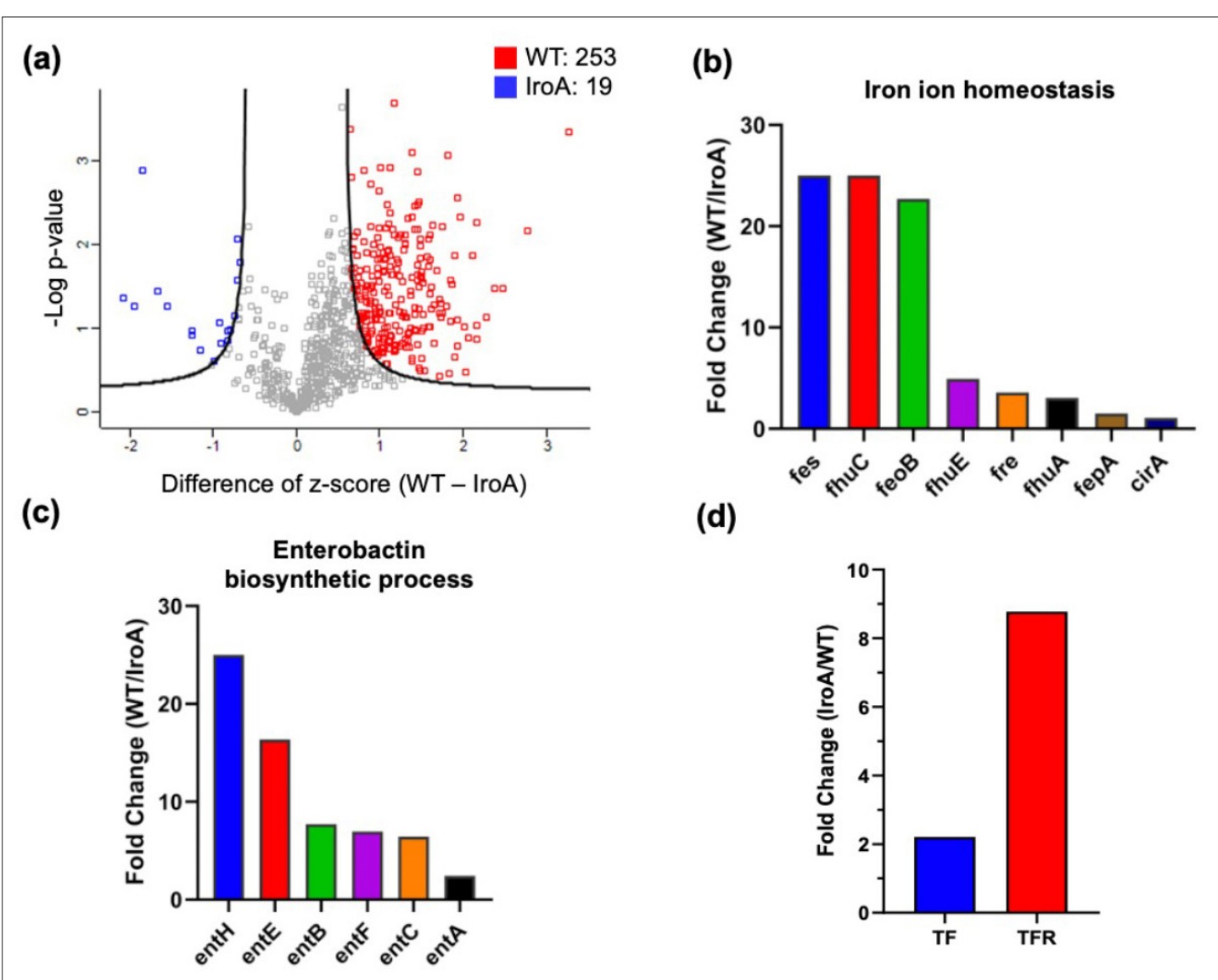

**Figure 4.** Quantitative proteomic analysis comparing IroA-*E. coli* and wild-type (WT)-*E. coli* in the tumor microenvironment (TME). (**a**) Volcano plot analysis between the proteomes of WT-*E. coli* and IroA-*E. coli* in the mouse tumors. (**b**) Fold changes of the proteins involved in the iron ion homeostasis. All the fold changes are >1. (**c**) Fold changes of the proteins involved in the enterobactin biosynthetic process. All the fold changes are >1. (**d**) Fold changes of transferrin and transferring receptor in the tumor.

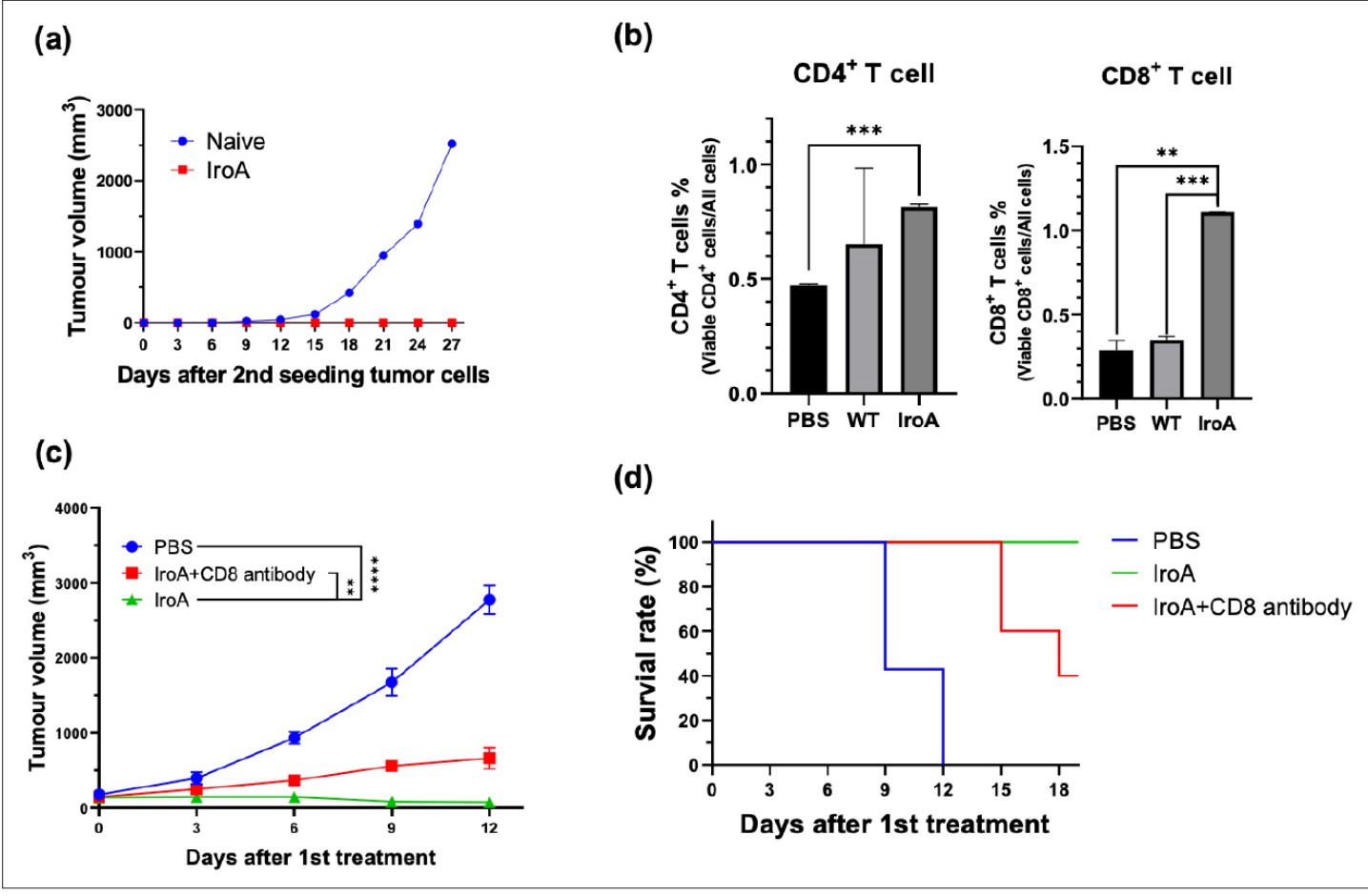

**Figure 5.** IroA-*E.coli* treatment stimulated the adaptive immune system for anti-tumor activity. (**a**) The mice cured by IroA-*E. coli* were re-challenged with a subcutaneous inoculation of 2.5×10⁵ MC38 cells. No tumor formation was observed. The naïve mice were used as controls. (**b**) The proportions of tumor-infiltrating CD4⁺ and CD8⁺ T cells in different treatment groups. (**c**) The tumor-bearing mice were treated with IroA-*E. coli* in the presence or absence of the anti-CD8 depletion antibody. (**d**) Survival curves of mice in different treatment groups. The error bars represent mean ± SD. Statistical analyses were performed by one-way ANOVA (**p<0.01, ***p<0.001).

The online version of this article includes the following figure supplement(s) for figure 5:

**Figure supplement 1.** IroA-*E. coli* treatment resulted in higher bacterial colonization in tumors as compared to wild-type (WT) bacteria.

treatment as compared to the WT-*E. coli* treatment (*Figure 4d*). These two proteins are known to be up-regulated when the cells sense a lack of iron in the environment (*Ponka and Lok, 1999*; *Theil, 1990*; *Ponka et al., 2015*). Our finding corroborates a competitive scenario between the host cells and *E. coli* where the potent acquisition of ferric ion by IroA-*E. coli* posed an iron-deficient stress to the host cells.

## Tumor suppression by iron-scavenging IroA-*E. coli* establishes anticancer adaptive immunity

Given that the IroA-*E. coli* treatment achieved complete tumor remission in multiple tumor models, we sought to investigate whether it also triggered adaptive immune responses against cancers. For the MC38-burdened mice that achieved complete tumor remission upon IroA-*E. coli* treatment, the mice were re-challenged with MC38 cancer cells 6 weeks following tumor eradication. The absence of tumor growth upon rechallenge indicates the establishment of robust anticancer adaptive immunity (*Figure 5a*), which led us to further analyze the tumor-infiltrating lymphocytes (TILs) in the mice treated by PBS, WT-*E. coli*, IroA-*E. coli*. We found that the TILs, especially the CD8⁺ T cells, were elevated in the IroA-*E. coli* group compared to the WT-*E. coli* group (*Figure 5b*), highlighting the superior capacity of IroA-*E. coli* treatment towards enhancing CD8⁺ T cell infiltration into the tumor

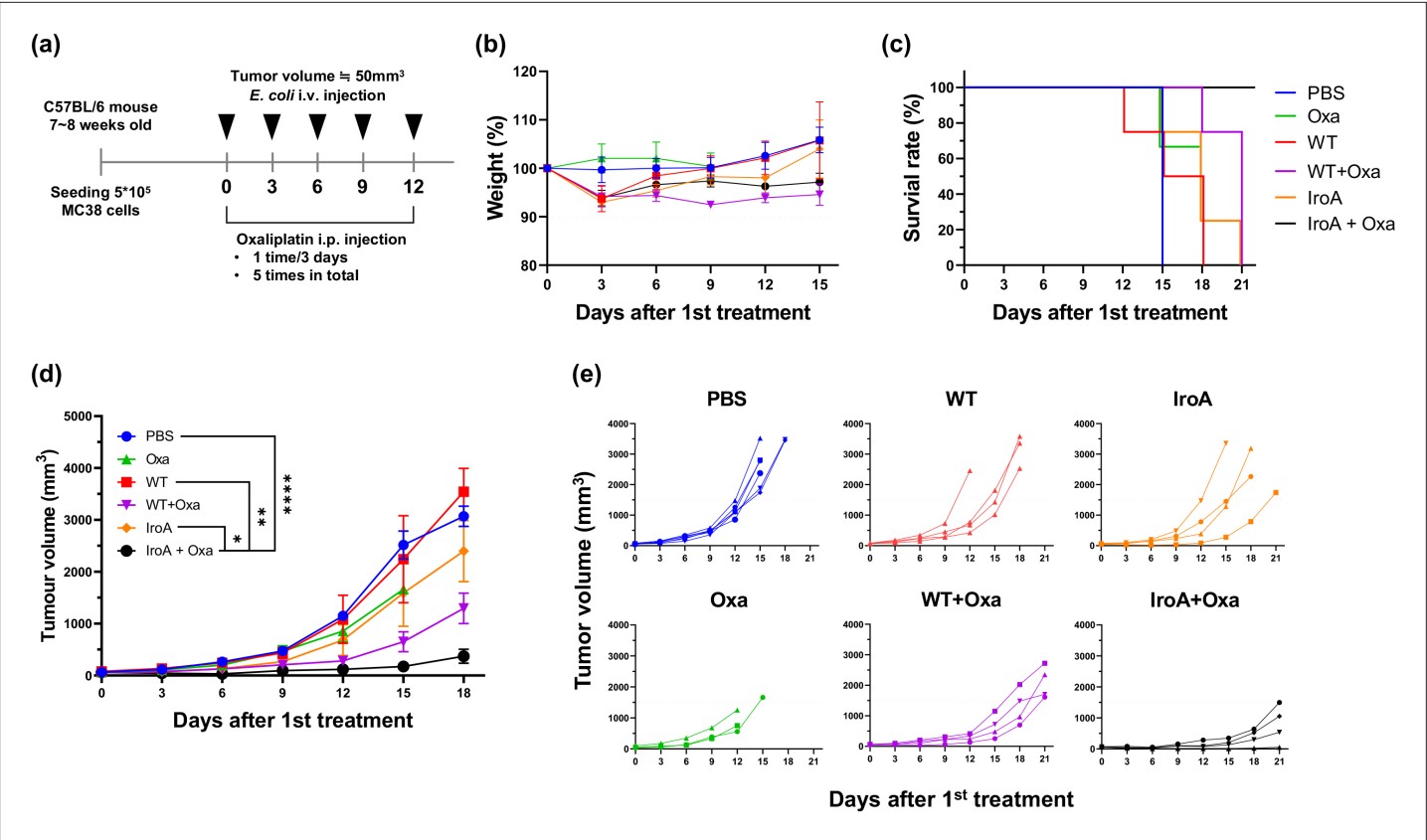

**Figure 6.** Synergistic anti-tumor activity of IroA-*E.coli* and oxaliplatin. (**a**) The scheme of the systemic delivery of IroA-*E. coli* and oxaliplatin in the tumor-bearing mice. (**b**) The alteration of mouse weights during the treatment course. (**c**) Survival curves of the mice in various treatment groups. (**d**) The average tumor growth curves of different treatment groups. (**e**) The tumor growth curves of individual mice in (**d**). The error bars represent mean ± SD. Statistical analyses were performed by one-way ANOVA (*$p<0.05$, **$p<0.01$, ****$p<0.0001$).

microenvironment. The enhanced CD8[+] T cell infiltration can be attributed to the prolonged intratumoral colonization by the IroA-*E. coli* (*Figure 5—figure supplement 1*), which can confer sustained immune stimulation for immune cell recruitment. We further performed a CD8[+] T-cell depletion study, which showed that the anti-tumor activity of IroA-*E. coli* was partially weakened upon anti-CD8 depletion (*Figure 5c*). These results demonstrate that the robust tumor suppression by the IroA-*E. coli* therapy can be attributed in part to the elicitation of tumor-specific cytotoxicity T cells, which can in turn provide durable anticancer immunity following bacterial clearance.

## Systemic delivery of IroA-*E. coli* and oxaliplatin show synergistic tumor suppression

It has been reported that *E. coli* possesses great tumor-targeting ability following intravenous injection in mice due to the hypoxic and immune-suppressive tumor microenvironment. Our previous study has also shown that the combination of *E. coli* and oxaliplatin synergistically suppresses the tumor growth (*Lim et al., 2024*). We, therefore, attempted to apply IroA-*E coli* and oxaliplatin for cancer treatment using a systemic delivery approach as depicted in *Figure 6a*. All mice maintained over 90% of their weights and remained active and healthy during the course of the treatments (*Figure 6b*). Unlike the intratumoral injection, the intravenous injection of *E. coli*, either the wild-type or the IroA transformant, resulted in subdued anti-tumor efficacy (*Figure 6c–e*). Also, the monotherapy of oxaliplatin only slightly inhibited the tumor growth. The combination of oxaliplatin with the wild-type *E. coli* delayed the tumor progression but did not achieve complete remission. Remarkably, the combination of oxaliplatin and IroA-*E. coli* significantly suppressed the tumor growth and achieved complete remission in 1 out of 4 mice. Overall, our data revealed that the systemic delivery of IroA-*E. coli* was synergistic with the oxaliplatin chemotherapy in the mouse tumor models.

## Discussion

There are accumulating studies that apply bacteria as drug delivery vehicles for cancer therapy with various payloads, including toxins, cytokines, immune checkpoint inhibitors, etc. However, in order to optimize the therapeutic outcomes of these engineering endeavors, it is also very important to understand the bacterial adaptation in the TME. The growth conditions differ vastly between the in vitro cultivation and the intratumoral environments. The rich medium provides ample nutrients for the optimal bacterial growth and payload production, whereas tumor is a nutrient-deprived, acidic, and hypoxic environment, which may stress the bacteria and restrain the engineered functionality. Also, unlike the in vitro cultivation, bacteria face the competition from the cancer cells as well as the surveillance from the host's immune system. This work provides the first proteomic data comparing *E. coli* cultured in the rich medium and colonized in the mouse tumor. The results show that the TME is an iron-deficient environment for *E. coli*. Moreover, upon the bacterial inoculation, the host cells up-regulate LCN2 to further block the iron uptake of *E. coli* by enterobactin (*Singh et al., 2015*; *Bachman et al., 2009*). This observation inspired us to develop *E. coli* that overcame the nutritional immunity from the host. This discovery-driven design, especially the IroA implementation, proves to be highly effective in enhancing the anti-tumor activity of *E. coli*. It is of interest to investigate that if the iron uptake ability is also critical for other types of bacteria when applied in cancer therapy.

In addition to the enterobactin biosynthetic process and the iron ion homeostasis, our proteomic data also revealed other insights into the key nutrients whose scarcity within the TME may impede the bacterial growth. For example, the proteins involving 'the de novo synthesis of IMP' are highly enriched in the *E. coli* grown in tumors compared to the rich medium condition. Inosine phosphate (IMP) is the precursor of purine. When the environment is deficient of purine, the bacteria need to synthesize them using de novo pathways (*Rolfes and Zalkin, 1988*; *Cho et al., 2011*; *Meng et al., 1990*). A previous report by Samant et al. has shown that the genes associated with the de novo purine synthesis are the most critical factors for the growth of *E. coli* or other gram-negative bacteria in human serum (*Samant et al., 2008*). Their data indicate that, similar to iron, humans also control the purine at a very low level in blood as a strategy of nutritional immunity in order to restrain the proliferation of bacteria. In light of these findings, one potential avenue could involve engineering *E. coli* to bolster its ability for de novo nucleotide biosynthesis, therefore, facilitating better adaptation to the TME. Given that bacteria and cancer cells vie for growth within the TME, an enhanced bacterial adaptation to the TME may potentially improve the anti-tumor activity.

Overall, our research revealed that the tug of war for iron plays a critical role when applying bacteria for cancer therapy. To aid the bacteria in this war, we have adopted several approaches, including the engineering of *E. coli* to secret cyclic-di-GMP and salmochelin. These methods have demonstrated effectiveness in treating murine tumors, resulting in a significant portion of complete remission. Notably, the cured mice have also established durable anti-tumor adaptive immunity. Our iron-scavenging strategy opens new avenues by overcoming the nutritional immunity hurdles in the realm of bacteria-based cancer therapy.

## Materials and methods

### Cancer cell strain and cultivation

The MC38 murine colon cancer cells were cultured in DMEM supplemented with 10% FBS, 1% penicillin-streptomycin solution (100 U/ml), 1 mM sodium pyruvate, 10mM HEPES, and 1% MEM Non-Essential Amino Acids Solution (100X). RAW264.7 macrophage cells were cultured in DMEM supplemented with 10%FBS, and 1% penicillin-streptomycin solution (100 U/ml). All cell lines were maintained in a humidified incubator at 37°C and 5% $CO_2$.

### Mouse experiments

All animal experiments were conducted under specific pathogen-free conditions according to the guidelines approved by the Animal Care and Usage Committee of Academia Sinica. Mice were housed at a temperature of 19–23°C with a 12-hr light-dark cycle and a humidity of 50–60%. A maximum of five mice were housed in a single individually ventilated cage with soft wood for nesting. Tumor dimensions were measured using a caliper, and tumor volume was calculated using the following formula: 0.52 x ((tumor length +tumor width)/2)$^3$. Mice aged 6–10 weeks were subcutaneously injected with

either $5\times10^5$ MC38 murine colon cancer cells, $5\times10^5$ E0771 breast cancer cells, or $1\times10^5$ B16F10 melanoma cancer cells suspended in 100 µL PBS at the right flank. The *Escherichia coli* strain BL21(DE3) was cultured in LB medium at 37°C overnight. The overnight cultures were diluted 100-fold in fresh LB medium and incubated at 37°C for ~3 hr until the log phage. Prior to the intratumoral injection, the bacteria density was determined by measuring the OD600 (1 OD = $4\times10^8$ CFU/ml). For the proteomic experiment, the tumor-bearing mice were intratumorally injected with $4\times10^8$ BL21(DE3) in 50 µL PBS when the tumor volumes reached approximately 150 mm$^3$. The tumors were harvested the following day for the LC-MS/MS analysis. For the intratumor-injection-based therapeutic experiment, the tumor-bearing mice were treated when the tumor volume reached ~150 mm$^3$. The tumor-bearing mice were intratumorally injected with $4\times10^8$ BL21(DE3) on day 0 and day 9. VLX 600 (4.5 mg/kg) was administered via intravenous injection every three days for a total of four injections. For the intravenous-injection-based therapeutic experiment, the tumor-bearing mice were intravenously injected with $1\times10^8$ BL21(DE3) in 100 µL PBS every three days for four times. The Oxaliplatin (5 mg/kg) was administered intraperitoneally every three days for a total of five injections. For the CD8$^+$ T cell depletion experiment, the mice received intraperitoneal injections of 100 µg anti-mouse CD8α antibodies (BioXCell, Cat# BE0061) on days −6,−2, 2, 6, and 10, along with the intratumoral injections of $4\times10^8$ BL21(DE3) on days 0 and 9. For tumor re-challenge study, MC38-burdened mice following bacterial therapy were subcutaneously injected with $5\times10^5$ MC38 murine colon cancer cells for monitoring. Naïve mice were challenged similarly as a control. Mice were considered dead and euthanized when the tumor reached a volume of 1500 mm$^3$. This study was performed in strict accordance with the recommendations in the Guide for the Care and Use of Laboratory Animals of the National Institutes of Health. All of the animals were handled according to approved institutional animal care and use committee (IACUC) protocols (Protocol#23-07-2027) of Academia Sinica.

## Proteomics sample preparation

For the tumor-colonized bacteria, the tumors were excised, and homogenized, and processed to extract intratumoral bacteria cells. Red blood cells were removed using RBC lysis buffer. The mouse cells were removed through low-speed centrifugation at 1200 g for 2 min three times. The *E coli* was collected by centrifugation at 4500 g for 20 min. The samples were lysed using 4% SDS, 100 mM Tris-HCL (pH 9), and 1 x protease inhibitor cocktail set III. The cell lysates were heated at 95°C for 5 min and sonicated for 15 min using a Bioruptor Plus (Diagenode). The supernatant was collected after centrifugation at 18,000 g for 30 min at 4°C. Approximately 50 µL supernatant was mixed with 200 µL methanol, 50 µL chloroform, and 150 µL double-distilled water. The aqueous phase was removed after sitting the sample at room temperature for 10 min. Subsequently, the sample was mixed with another 150 µL methanol. The pellet was collected, dried for 20 min, and resuspended in 8 M urea and 50 mM triethylammonium bicarbonate buffer. The samples were reduced with 10 mM DTT, alkylated with 50 mM IAA, and digested using LysC and trypsin. Following acidification, the supernatant was loaded onto the SDB-XC StageTips (*Rappsilber et al., 2007*) and eluted by 80% ACN containing 0.1%TFA. The sample was lyophilized and stored at −20°C before further LC-MS/MS analysis. For the bacteria grown in the rich medium, the *E. coli* strain BL21(DE3) was cultured in LB medium at 37°C overnight. The overnight culture was diluted 100-fold in fresh LB medium and incubated at 37°C until the log phage (OD = 0.6). The bacteria were harvested by centrifugation and treated similarly to the bacteria harvested from tumors.

## LC-MS/MS experiments

The sample was loaded onto the trap column (2 cm ×75 µm i.d., Symmetry C18), and then separated on a nanoACQUITY UPLC System (Waters, USA) equipped with a 25 cm ×75 µm i.d. BEH130 C18 column (Waters, USA) using a 5–35% buffer B (buffer A: 0.1% formic acid; buffer B: 0.1% formic acid in acetonitrile) gradient as the separation phase and a flow rate of 300 nl/min. The total running time was 120 min. The mass spectrometric data were collected on a high-resolution Q Exactive HF-X mass spectrometer (Thermo Fisher Scientific, Bremen, Germany) operating in the data-dependent mode. Full MS resolution was set to 60,000 at 200 m/z and the mass range was set to 350–1600. dd-MS2 resolution was set to 15,000 at 200 m/z. Isolation width was set to 1.3 m/z. Normalized collision energy was set to 28%. The LC-MS/MS data were matched with the human SwissProt database using

the Mascot search engine v.2.6.1 (Matrix Science, UK) with the following parameters: the mass tolerance of precursor peptide was set to 10 ppm, and the tolerance for MS/MS fragments was 0.02 Da.

## Proteomics data processing and statistical analysis

Raw MS data were processed using MaxQuant version 2.0.1. Database search was performed with the Andromeda search engine through the Uniprot database (*Cox et al., 2011*). Both protein and peptide levels were filtered by a 1% false discovery rate (FDR). The variable modification setting included oxidation(M) and Acetyl (Protein N-term), and the fixed modification setting included carbamidomethyl(C). The 'match between runs' was set as 1 min, and the MaxQuant LFQ algorithm was employed for normalization. The statistical analysis was performed using Perseus version 1.6.15.0 and Prism version 8.0.2 (*Tyanova et al., 2016*). The proteinGroups output table from MaxQuant was utilized for proteomics analysis. The potential contaminant, reverse, and only-identified-by-site were filtered out. The LFQ intensity was log2-transformed and filtered for validity. NaN values were imputed, and bacterial cell LFQ intensity was normalized using a z-score. The LFQ intensity of bacterial cells was normalized using a z-score (n-average/standard deviation). A t-test with an FDR of 0.05 and S0 of 1 was performed to extract significantly different proteins. These proteins were uploaded to the DAVID database for biological interpretation, and the results were visualized in Prism. The raw data of LFQ intensity for significantly different proteins were averaged to calculate the difference in protein expression level.

## Enterobactin extraction

BL21(DE3) was cultured in the LB medium at 37°C overnight. The overnight culture was 100-fold diluted to the M9 medium supplemented with 0.2% casamino acids, 0.2% glucose, 1 mM $MgSO_4$, and 1 mg/mL vitamin B1 and grown for 20 hr. The bacteria were removed by centrifugation, and the supernatant was sterilized using a 0.22 μm filter. The cell-free supernatant was acidified to pH = 2 using 10 N HCl. An equal volume of ethyl acetate was added to the acidified supernatant and mixed using Intelli Mixer ERM-2L. The organic fraction was collected after 30 min incubation at room temperature and dried using a miVac centrifugal concentrator. All samples were resuspended in DMSO and stored at –20°C for further experiments.

## Cytotoxicity assay of enterobactin

$1 \times 10^4$ MC38 cells in a 96-well plate were treated with the enterobactin extracted from WT-*E. coli* or IroA-*E. coli* for 48 hr. The cell viability was measured using the Cell Counting Kit-8 (CCK-8) according to the manufacturer's protocol. A cell-free mixture was used as a background reference, and the untreated cells were used as a control.

## Characterization of cyclic-di-GMP (CDG) secreted from DGC-*E. coli*

The DGC plasmid carries the gene of the diguanylate cyclase fragment 82–248 residues from *Thermotoga maritima* with a single mutation of Arg158Ala. The BL21(DE3) bacteria transformed with the DGC plasmid were cultured in LB at 37°C overnight. The overnight cultures were 100-fold diluted to fresh LB supplemented with 0.1 mM IPTG and kanamycin (50 ug/ml) and cultured at 37°C for 20 hr. The bacteria were pelleted, and the supernatant was collected and filtered using a 0.22 μm strainer. The supernatant was analyzed by LC-MS/MS to identify the CDG.

## Interferon-β quantification

$5 \times 10^5$ RAW264.7 cells in a 24-well plate were treated with the conditioned medium from DGC-*E. coli* or non-transformed *E. coli* for 18 hr. Subsequently, the cell culture medium was collected for IFN-β quantification. The IFN-β levels were measured using the Mouse IFN-beta ELISA kit (R&D, #P318019) following the manufacturer's protocol.

## Iron chelating assay

The iron uptake ability of bacteria was determined using chrome azurol S (CAS) assay (*Louden et al., 2011*). The BL21(DE3) cells with or without IroA transformation were cultured in the M9 medium at 37°C overnight. The bacteria were collected by centrifugation and washed twice with PBS. $10^8$ bacteria were inoculated into fresh M9 medium with CAS reagent and different concentrations of Lcn2

protein and incubated at 37°C overnight. The A630 of the bacterial cultured medium was measured to quantify the iron-consuming ability of the bacteria.

## Lipocalin 2 resistance assay

The IroA-*E. coli* or non-transformed *E. coli* was cultured overnight in the LB medium. Next day, the bacterial culture was diluted to fresh RPMI supplemented with 10% FBS by 100-fold and incubated at 37°C for 5 hr to reach the log phase. $10^5$ bacteria were treated with different concentrations of LCN2 and incubated at 37°C for 20 hr. The live bacteria were quantified by serial tittering on LB agar plates.

## Tumor-infiltrating lymphocyte (TIL) analysis

Tumor tissues were cut into small pieces in the digestion buffer (1 mL RPMI supplemented with 10% FBS, 0.33 mg collagenase type IV (Sigma, Cat#C5138), and 66 µg DNase I from bovine pancreas (Cyrusbiosicence, Cat#101-9003-98-9)) and transferred into a C tube followed by sample processing using gentleMACS. The cell suspensions were filtered through a 40 µm strainer. The RBC lysis buffer was added to the tumor suspension to remove red blood cells. The cells were then blocked using the CD16/32 Fc blocker for 5 min on ice. The T cells were stained with eFluor780 viability dye, PE-CD45, APC-CD4, and FITC-CD8a antibodies. The stained T cells were analyzed by flow cytometer (Attune NxT cytometer), and the data were processed using FlowJo software.

## Antibodies

| Antibodies | Supplier | Catalog number |
|---|---|---|
| anti-mouse CD16/32 Antibody | Biolegend | 101301 |
| PE anti-mouse CD45 Antibody | Biolegend | 103105 |
| APC anti-mouse CD4 Antibody | Biolegend | 100516 |
| FITC anti-mouse CD8a Antibody | Biolegend | 100706 |
| InVivoMAb anti-mouse CD8α | Bio X Cell | BE0061 |

## Acknowledgements

We thank Academia Sinica Core Facility and Innovative Instrument Project (AS-CFII-111–212). We thank the Academia Sinica DNA Sequencing Core Facility (AS-CFII-108–115). This work was supported by an Academia Sinica Career Development Award (AS-CDA-108-L07) and the Ministry of Science and Technology, Taiwan (110–2113 M-001-064-MY3).

## Additional information

### Funding

| Funder | Grant reference number | Author |
|---|---|---|
| Academia Sinica | AS-CDA-108-L07 | Kurt Yun Mou |
| National Science and Technology Council | 110-2113-M-001-064-MY3 | Kurt Yun Mou |

The funders had no role in study design, data collection and interpretation, or the decision to submit the work for publication.

### Author contributions

Sin-Wei Huang, Conceptualization, Data curation, Formal analysis, Validation, Investigation, Methodology, Writing – original draft, Writing – review and editing; See-Khai Lim, Data curation, Formal analysis, Investigation; Yao-An Yu, Yi-Chung Pan, Data curation, Investigation; Wan-Ju Lien, Data curation, Validation, Investigation; Chung-Yuan Mou, Supervision, Writing – review and editing; Che-Ming Jack Hu, Resources, Supervision, Methodology, Project administration, Writing – review and editing; Kurt

Yun Mou, Conceptualization, Resources, Formal analysis, Writing – original draft, The author passed away on August 28th, 2023

## Author ORCIDs
Sin-Wei Huang http://orcid.org/0009-0004-1593-6511
Che-Ming Jack Hu https://orcid.org/0000-0002-0988-7029
Kurt Yun Mou https://orcid.org/0000-0001-5423-9031

## Ethics
This study was performed in strict accordance with the recommendations in the Guide for the Care and Use of Laboratory Animals of the National Institutes of Health. All of the animals were handled according to approved institutional animal care and use committee (IACUC) protocols (Protocol#23-07-2027) of Academia Sinica.

Reviewer #1 (Public Review): https://doi.org/10.7554/eLife.90798.3.sa1
Reviewer #2 (Public Review): https://doi.org/10.7554/eLife.90798.3.sa2
Reviewer #3 (Public Review): https://doi.org/10.7554/eLife.90798.3.sa3
Author response https://doi.org/10.7554/eLife.90798.3.sa4

---

# Additional files

## Supplementary files
• MDAR checklist
• Source data 1. Source data for figure preparation.

## Data availability
All data generated or analysed during this study are included in the manuscript and supporting files. Proteomics data can be accessed at Dryad (https://doi.org/10.5061/dryad.z08kprrnn). Source data used for the plots in the study is included in *Source data 1*.

The following dataset was generated:

| Author(s) | Year | Dataset title | Dataset URL | Database and Identifier |
|---|---|---|---|---|
| CHJ Hu | 2024 | Data from: Overcoming the nutritional immunity by engineering iron scavenging bacteria for cancer therapy | http://dx.doi.org/10.5061/dryad.z08kprrnn | Dryad Digital Repository, 10.5061/dryad.z08kprrnn |

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
