## [Editor Report · eLife assessment]

This **valuable** study combines proteomics and a mouse model to reveal the importance of iron uptake in bacterial therapy for cancer. The evidence presented is **convincing**. Notably, the authors showed upregulation of iron uptake of bacteria significantly inhibits tumor growth in vivo. This paper will be of interest to a broad audience including researchers in cancer biology, cell biology, and microbiology.

---

## [Referee Report · Reviewer #1 (Public Review)]

In this manuscript, Huang and colleagues explored the role of iron in bacterial therapy for cancer. Using proteomics, they revealed the upregulation of bacterial genes that uptake iron, and reasoned that such regulation is an adaptation to the iron-deficient tumor microenvironment. Logically, they engineered *E. coli* strains with enhanced iron-uptake efficiency, and showed that these strains, together with iron scavengers, suppress tumor growth in a mouse model. Lastly, they reported the tumor suppression by IroA-*E. coli* provides immunological memory via CD8+ T cells. In general, I find the findings in the manuscript novel and the evidence convincing.

(1) Although the genetic and proteomic data are convincing, would it be possible to directly quantify the iron concentration in (1) *E. coli* in different growth environments, and (2) tumor microenvironment? This will provide functional consequence of upregulating genes that import iron into the bacteria.

(2) Related to 1, the experiment to study the synergistic effect of CDG and VLX600 (lines 139-175) is very nice and promising, but one flaw here is a lack of the measurement of iron concentration. Therefore, a possible explanation could be that CDG acts in another manner, unrelated to iron uptake, that synergizes with VLX600's function to deplete iron from cancer cells. Here, a direct measurement of iron concentration will show the effect of CDG on iron uptake, thus complementing the missing link.

(3) Lines 250-268: Although statistically significant, I would recommend the authors characterize the CD8+ T cells a little more, as the mechanism now seems quite elusive. What signals or memories do CD8+ T cells acquire after IroA-*E. coli* treatment to confer their long-term immunogenicity?

(4) Perhaps this goes beyond the scope of the current manuscript, but how broadly applicable is the observed iron-transport phenomenon in other tumor models? I would recommend the authors to either experimentally test it in another model, or at least discuss this question.

---

## [Referee Report · Reviewer #2 (Public Review)]

Summary:

The authors provide strong evidence that bacteria, such as *E. coli*, compete with tumor cells for iron resources and consequently reduce tumor growth. When sequestration between LCN2 and bacterobactin is blocked by upregulating CDG(DGC-*E. coli*) or salmochelin(IroA-E.coli), *E. coli* increase iron uptake from the tumor microenvironment (TME) and restrict iron availability for tumor cells. Long-term remission in IroA-E.coli treated mice is associated with enhanced CD8+ T cell activity. Additionally, systemic delivery of IroA-E.coli shows a synergistic effect with chemotherapy reagent oxaliplatin to reduce tumor growth.

Strengths:

It is important to identify the iron-related crosstalk between *E. coli* and TME. Blocking lcn2-bacterobactin sequestration by different strategies consistently reduce tumor growth.

Weaknesses:

As engineered *E. coli* upregulate their function to uptake iron, they may increase the likelihood of escaping from nutritional immunity (LCN2 becomes insensitive to sequester iron from the bacteria). Would this raise the chance of developing sepsis? Do authors think that it is safe to administrate these engineered bacteria in mice or humans?

---

## [Referee Report · Reviewer #3 (Public Review)]

Summary:

Based on their observation that tumor has an iron-deficient microenvironment, and the assumption that nutritional immunity is important in bacteria-mediated tumor modulation, the authors postulate that manipulation of iron homeostasis can affect tumor growth. This paper uses straightforward in vitro and in vivo techniques to examine a specific and important question of nutritional immunity in bacteria-mediated tumor therapy. They are successful in showing that manipulation of iron regulation during nutritional immunity does affect the virulence of the bacteria, and in turn the tumor. These findings open future avenues of investigation, including the use of different bacteria, different delivery systems for therapeutics, and different tumor types. The authors were also successful in addressing the reviewer's concerns adequately.

---

## [Author Response]

The following is the authors’ response to the previous reviews.

**Reviewer #1 (Public Review):**
In this manuscript, Huang and colleagues explored the role of iron in bacterial therapy for cancer. Using proteomics, they revealed the upregulation of bacterial genes that uptake iron, and reasoned that such regulation is an adaptation to the iron-deficient tumor microenvironment. Logically, they engineered *E. coli* strains with enhanced iron-uptake efficiency, and showed that these strains, together with iron scavengers, suppress tumor growth in a mouse model. Lastly, they reported the tumor suppression by IroA-*E. coli* provides immunological memory via CD8+ T cells. In general, I find the findings in the manuscript novel and the evidence convincing.(1) Although the genetic and proteomic data are convincing, would it be possible to directly quantify the iron concentration in (1) *E. coli* in different growth environments, and (2) tumor microenvironment? This will provide the functional consequences of upregulating genes that import iron into the bacteria.

We appreciate the reviewer’s comment regarding the precise quantification of iron concentrations. In our study, we attempted various experimental approaches, including Immunohistochemistry utilizing an a Fe3+ probe, iron assay kit (ab83366), and Inductively Coupled Plasma Mass Spectrometry (ICP-MS). Despite these attempts, the quantification of oxidized Fe3+ concentrations proved challenging due to the inherently low levels of Fe ions and difficulty to distinguish Fe2+ and Fe3+. We observed measurements below the detection threshold of even the sensitive ICP-MS technique. To circumvent this limitation, we designed an experiment wherein bacteria were cultured in a medium supplemented with Chrome Azurol S (CAS) reagent, which colormetrically detects siderophore activity. We compared WT bacteria and IroA-expressing bacteria at varying levels of Lcn2 proteins. The outcome, as depicted in the updated Fig. 3b, reveals an enhanced iron acquisition capability in IroA-*E. coli* under the presence of Lcn2 proteins, in comparison to the wild-type *E. coli* strains. In addition to the Lcn2 study, the proteomic study in Figure 4 highlights the competitive landscape between cancer cells and bacteria. We observed that IroA-E. coli showed reduced stress responses and exerted elevated iron-associated stress to cancer cells, thus further supporting the IroA-*E. coli*’s iron-scavenging capability against nutritional immunity.

(2) Related to 1, the experiment to study the synergistic effect of CDG and VLX600 (lines 139-175) is very nice and promising, but one flaw here is a lack of the measurement of iron concentration. Therefore, a possible explanation could be that CDG acts in another manner, unrelated to iron uptake, that synergizes with VLX600's function to deplete iron from cancer cells. Here, a direct measurement of iron concentration will show the effect of CDG on iron uptake, thus complementing the missing link.

We appreciate the reviewer’s comment and would like to point the reviewer to our results in Figure S3, which shows that the expression of CDG enhances bacteria survival in the presence of LCN2 proteins, which reflects the competitive relationship between CDG and enterobactin for LCN2 proteins as previously shown by Li et al. [Nat Commun 6:8330, 2015]. We regret to inform the reviewer that direct measurement of iron concentration was attempted to no avail due to the limited sensitivity of iron detecting assays. We do acknowledge that CDG may exert different effects in addition to enhancing iron uptake, particularly the potentiation of the STING pathway. We pointed out such effect in Fig 2c that shows enhanced macrophage stimulation by the CDG-expressing bacteria. We would like to accentuate, however, that a primary objective of the experiment is to show that the manipulation of nutritional immunity for promoting anticancer bacterial therapy can be achieved by combining bacteria with iron chelator VLX600. The multifaceted effects of CDG prompted us to focus on IroA-*E. coli* in subsequent experiments to examine the role of nutritional immunity on bacterial therapy. We have updated the associated text to better convey our experimental design principle.

Lines 250-268: Although statistically significant, I would recommend the authors characterize the CD8+ T cells a little more, as the mechanism now seems quite elusive. What signals or memories do CD8+ T cells acquire after IroA-*E. coli* treatment to confer their long-term immunogenicity?

We apologize for the overinterpretation of the immune memory response in our previous manuscript and appreciate the reviewer’s recommendation to further characterize CD8+ T cells post-IroA-*E. coli* treatment. Our findings, which show robust tumor inhibition in rechallenge studies, indicate establishment of anticancer adaptive immune responses. As the scope of the present work is aimed at demonstrating the value of engineered bacteria for overcoming nutritional immunity, expounding on the memory phenotypes of the resulting cellular immunity is beyond the scope of the study. We do acknowledge that our initial writing overextended our claims and have revised the manuscript accordingly. The revised manuscript highlights induction of anticancer adaptive immunity, attributable to CD8+ T cells, following the bacterial therapy.

(3) Perhaps this goes beyond the scope of the current manuscript, but how broadly applicable is the observed iron-transport phenomenon in other tumor models? I would recommend the authors to either experimentally test it in another model or at least discuss this question.

We highly appreciate the reviewer’s suggestion regarding the generalizability of the iron-transport phenomenon in diverse tumor models. To address this, we extended our investigations beyond the initial model, employing B16-F10 melanoma and E0771 breast cancer in mouse subcutaneous models. The results, as depicted in Figures 3g to 3j and Figure S5, demonstrate the superiority of IroA-*E. coli* over WT bacteria in tumor inhibition. These findings support the broad implication of nutritional immunity as well as the potential of iron-scavenging bacteria for different solid tumor treatments.

**Reviewer #2 (Public Review):**
Summary:The authors provide strong evidence that bacteria, such as *E. coli*, compete with tumor cells for iron resources and consequently reduce tumor growth. When sequestration between LCN2 and bacterobactin is blocked by upregulating CDG(DGC-*E. coli*) or salmochelin(IroA-E.coli), *E. coli* increase iron uptake from the tumor microenvironment (TME) and restrict iron availability for tumor cells. Long-term remission in IroA-E.coli treated mice is associated with enhanced CD8+ T cell activity. Additionally, systemic delivery of IroA-E.coli shows a synergistic effect with chemotherapy reagent oxaliplatin to reduce tumor growth.Strengths:It is important to identify the iron-related crosstalk between *E. coli* and TME. Blocking lcn2-bacterobactin sequestration by different strategies consistently reduces tumor growth.Weaknesses:As engineered *E. coli* upregulate their function to uptake iron, they may increase the likelihood of escaping from nutritional immunity (LCN2 becomes insensitive to sequester iron from the bacteria). Would this raise the chance of developing sepsis? Do authors think that it is safe to administrate these engineered bacteria in mice or humans?

We appreciate the reviewer’s comment on the safety evaluation of the iron-scavenging bacteria. To address the concern, we assessed the potential risk of sepsis development by measuring the bacterial burden and performing whole blood cell analyses following intravenous injection of the engineered bacteria. As illustrated in Figures 3k and 3l, our findings indicate that the administration of these engineered bacteria does not elevate the risk of sepsis. The blood cell analysis suggests that mice treated with the bacteria eventually return to baseline levels comparable to untreated mice, supporting the safety of this approach in our experimental models.

**Reviewer #3 (Public Review):**
Summary:Based on their observation that tumor has an iron-deficient microenvironment, and the assumption that nutritional immunity is important in bacteria-mediated tumor modulation, the authors postulate that manipulation of iron homeostasis can affect tumor growth. They show that iron chelation and engineered DGC-*E. coli* have synergistic effects on tumor growth suppression. Using engineered IroA-*E. coli* that presumably have more resistance to LCN2, they show improved tumor suppression and survival rate. They also conclude that the IroA-*E. coli* treated mice develop immunological memory, as they are resistant to repeat tumor injections, and these effects are mediated by CD8+ T cells. Finally, they show synergistic effects of IroA-*E. coli* and oxaliplatin in tumor suppression, which may have important clinical implications.Strengths:This paper uses straightforward in vitro and in vivo techniques to examine a specific and important question of nutritional immunity in bacteria-mediated tumor therapy. They are successful in showing that manipulation of iron regulation during nutritional immunity does affect the virulence of the bacteria, and in turn the tumor. These findings open future avenues of investigation, including the use of different bacteria, different delivery systems for therapeutics, and different tumor types.Weaknesses:There is no discussion of the cancer type and why this cancer type was chosen. Colon cancer is not one of the more prominently studied cancer types for LCN2 activity. While this is a proof-of-concept paper, there should be some recognition of the potential different effects on different tumor types. For example, this model is dependent on significant LCN production, and different tumors have variable levels of LCN expression. Would the response of the tumor depend on the role of iron in that cancer type? For example, breast cancer aggressiveness has been shown to be influenced by FPN levels and labile iron pools.

We highly appreciate the reviewer’s insightful comment on the varying LCN2 activities across different tumor types. In light of the reviewer’s suggestion, we extended our investigations beyond the initial colon cancer model, employing B16-F10 melanoma and E0771 breast cancer in mouse subcutaneous models. The results, as depicted in Figures 3g to 3j and Figure S5, demonstrate that IroA-*E. coli* consistently outperforms WT bacteria in tumor inhibition. We acknowledge the reviewer’s comment regarding LCN2 being more prominently examined in breast cancer and have highlighted this aspect in the revised manuscript. For colon and melanoma cancers, several reports have pointed out the correlation of LCN2 expression and the aggressiveness of these cancers [Int J Cancer. 2021 Oct 1;149(7):1495-1511][Nat Cancer. 2023 Mar;4(3):401-418], albeit to a lesser extent. These findings support the broad implication of nutritional immunity as well as the potential of iron-scavenging bacteria for different solid tumor treatments. The manuscript has been revised to reflect the reviewer’s insightful comment.

Are the effects on tumor suppression assumed to be from *E. coli* virulence, i.e. Does the higher number of bacteria result in increased immune-mediated tumor suppression? Or are the effects partially from iron status in the tumor cells and the TME?

We appreciate the reviewer’s question regarding the therapeutic mechanism of IroA*-E. coli*. Bacterial therapy exerts its anticancer action through several different mechanisms, including bacterial virulence, nutrient and ecological competition, and immune stimulation. Decoupling one mechanism from another would be technically challenging and beyond the scope of the present work. With the objective of demonstrating that an iron-scavenging bacteria can elevate anticancer activity by circumventing nutritional immunity, we highlight our data in Fig. S6, which shows that IroA-*E. coli* administration resulted in higher bacterial colonization within solid tumors compared to WT-*E. coli* on Day 15. This increased bacterial presence supports our iron-scavenging bacteria design, and we highlight a few anticancer mechanisms mediated by the engineered bacteria. Firstly, as shown in Fig. 4d, IroA-*E. coli* is shown to induce an elevated iron stress response in tumor cells as the treated tumor cells show increased expression of transferrin receptors. Secondly, our experiments involving CD8+ T cell depletion indicates that the IroA-*E. coli* establishes a more robust anticancer CD8+ T cell response than WT bacteria. Both immune-mediated responses and alterations in iron status within the tumor microenvironment are demonstrated to contribute to the enhanced anticancer activity of IroA*-E. coli* in the present study.

If the effects are iron-related, could the authors provide some quantification of iron status in tumor cells and/or the TME? Could the proteomic data be queried for this data?

We appreciate the reviewer’s query regarding the quantification of iron concentrations. In our study, we attempted various experimental approaches, including Immunohistochemistry utilizing an a Fe3+ probe, iron assay kit (ab83366), and Inductively Coupled Plasma Mass Spectrometry (ICP-MS). Despite these attempts, the quantification of oxidized Fe3+ concentrations proved challenging due to the inherently low levels of Fe ions and difficulty to distinguish Fe2+ and Fe3+. We observed measurements below the detection threshold of even the sensitive ICP-MS technique. Consequently, to circumvent this limitation, we designed an experiment wherein bacteria were cultured in a medium supplemented with Chrome Azurol S (CAS) reagent, which colormetrically detects siderophore activity. We compared WT bacteria and IroA-expressing bacteria at varying levels of Lcn2 proteins. The outcome, as depicted in the updated Fig. 3b, reveals an enhanced iron acquisition capability in IroA-*E. coli* under the presence of Lcn2 proteins, in comparison to the wild-type *E. coli* strains. In addition to the Lcn2 study, the proteomic study in Figure 4 highlights the competitive landscape between cancer cells and bacteria. We observed that IroA-*E. coli* showed reduced stress responses and exerted elevated iron-associated stress to cancer cells, thus further supporting the IroA-*E. coli’s* iron-scavenging capability against nutritional immunity.

**Reviewing Editor:**
The authors provide compelling technically sound evidence that bacteria, such as *E. coli*, can be engineered to sequester iron to potentially compete with tumor cells for iron resources and consequently reduce tumor growth. Long-term remission in IroA-E.coli treated mice is associated with enhanced CD8+ T cell activity and a synergistic effect with chemotherapy reagent oxaliplatin is observed to reduce tumor growth. The following additional assessments are needed to fully evaluate the current work for completeness; please see individual reviews for further details.

We appreciate the editor’s positive comment.

(1) The premise is one of translation yet the authors have not demonstrated that manipulating bacteria to sequester iron does not provide a potential for sepsis or other evidence that this does not increase the competitiveness of bacteria relative to the host. Only tumor volume was provided rather than animal survival and cause of death, but bacterial virulence is enhanced including the possibility of septic demise. Alternatively, postulated by the authors, that tumor volume is decreased due to iron sequestration but they do not directly quantify the iron concentration in (1) *E. coli* in different growth environments, and (2) tumor microenvironment. These important endpoints will provide the functional consequences of upregulating genes that import iron into the bacteria.

We appreciate the editor’s comment and have added substantial data to support the translational potential of the iron-scavenging bacteria. In particular, we added evidence that the iron-scavenging bacteria does not increase the risk of sepsis (Fig. 3k, l), evidence of increased bacteria competitiveness and survival in tumor (Fig. S6), and iron-scavenging bacteria’s superior anticancer ability and survival benefit across 3 different tumor models (Fig. 3e-j; Fig. S5). While direct measurement of iron concentration in the tumor environment is technically difficult due to the challenge in differentiating Fe2+ and Fe3+ by available techniques, we added a colormetric CAS assay to demonstrate the iron-scavenging bacteria can more effectively utility Fe than WT bacteria in the presence of LCN2 (Fig. 3b). These results substantiate the translational relevance of the engineered bacteria.

(2) There is no discussion of the cancer type and why this cancer type was chosen. If the current tumor modulation system is dependent on LCN2 activity, there would need to be some recognition that different tumors have variable levels of LCN expression. Would the response of the tumor depend on the role of iron in that cancer type?

We appreciate the comment and added relevant text and citations describing clinical relevance of LCN2 expression associated with the tumor types used in the study (breast cancer, melanoma, and colon cancer). Elevated LCN2 has been associated with higher aggressiveness for all three cancer types.

(3) To demonstrate long-term anti-cancer memory was established through enhancement of CD8+ T cell activity (Fig 5c), the "2nd seeding tumor cells" experiment may need to be done in CD8 antibody-treated IronA mice since CD8+ T cells may play a role in tumor suppression regardless of whether or not iron regulation is being manipulated. It appears that the control group for this experiment is naive mice (and not WT-*E. coli* treated mice), in which case the immunologic memory could be from having had tumor/*E. coli* rather than the effect of IroA-*E. coli.*

We acknowledge that our prior writing may have overstated our claim on immunological memory. Our intention is to show that upon treatment and tumor eradication by iron-scavenging bacteria, adaptive immunity mediated by CD8 T cells can be elicited. We also did not consider a WT-*E. coli* control as no WT-*E. coli* treated group achieved complete tumor regression. We have modified our text to reflect our intended message.

**Reviewer #1 (Recommendations For The Authors):**
All the figures seem to be in low resolution and pixelated. Please upload high-resolution ones.

We have updated figures to high-resolution ones.

**Reviewer #2 (Recommendations For The Authors):**
Some specific comments towards experiments:(1) For Fig 2 f/ Fig 3f/ Fig 5d/Fig6c, the survival rate is based on the tumor volume (the mouse was considered dead when the tumor volume exceeded 1,500 mm3). Did the mice die from the experiment (how many from each group)? If it only reflects the tumor size, do these figures deliver the same information as the tumor growth figure?

We appreciate the reviewer’s comment. The survival rate is indeed based on tumor volume, and we used a cutoff of 1500 mm3. No death event was observed prior to the tumors reaching 1500 mm3. Although the survival figures cover some of the information conveyed by the tumor volume tracking, the figures offer additional temporal resolution of tumor progression with the survival figures. Having both tumor volume and survival tracking are commonly adopted to depict tumor progression. We have the protocol regarding survival monitoring to the materials and method section.

(2) Fig 3a, not sure if entE is a good negative control for this experiment. Neg. Ctrl should maintain its CFU/ml at a certain level regardless of Lcn2 conc. However, entE conc. is at 100 CUF/ml throughout the experiment suggesting there is no entE in media or if it is supersensitive to Lcn2 that bacteria die at the dose of 0.1nM?

We appreciate the reviewer’s comment. The △entE-*E. coli* was indeed observed to be highly sensitive to LCN2. We included the control to highlight the competitive relationship between entE and LCN2 for iron chelation, which is previously reported in literature [Biometals 32, 453–467 (2019)].

(3) Fig 4, the authors harvested bacteria from the tumor by centrifuging homogenized samples at different speeds. Internal controls confirming sample purity (positive for bacteria and negative for cells for panels a,b,c; or vice versa for panel d) may be necessary. This comment may also apply to samples from Fig 1.

We acknowledge the reviewer’s concern and would like to point out that the proteomic analysis was performed using a highly cited protocol that provides reference and normalization standards for *E. coli* proteins [Mol Cell Proteomics. 2014 Sep; 13(9): 2513–2526]. The reference is cited in the Materials and Method section associated with the proteomic analysis.

(4) To demonstrate long-term anti-caner memory was established through enhancement of CD8+ T cell activity, the "2nd seeding tumor cells" experiment may need to be done in CD8 antibody-treated IronA mice.

We have modified our claims to highlight that the tumor eradication by iron scavenging bacteria can establish adaptive anticancer immunity through the elicitation of CD8 T cells. We apologize for overstating our claim in the previous manuscript draft.

Minor suggestions:(1) Please include the tumor re-challenge experiment in the method section.

The re-challenge experiment has been added to the method section as instructed.

(2) Please cite others' and your previous work. E.g. line 281, 282, line 306-307.

We have added the citations as instructed.

(3) Line 448, BL21 is bacteria, not cells.

We have made the correction accordingly.

**Reviewer #3 (Recommendations For The Authors):**
The authors postulate that IroA-*E. coli* is more potent than DGC-*E. coli* in resisting LCN2 activity, and that this potency is the cause of the increased tumor suppression of this engineered strain. If so, Fig 3a should include DGC-*E. coli* for direct comparison.

We appreciate the reviewer for the comment and would like to clarify that we intended construct IroA-*E. coli* as a more specific iron-scavenging strategy, which can aide the discussion of nutritional immunity and minimize compounding factors from the immune-stimulatory effect of CDG. We have modified our text to clarify our stance.

The data refers to the effects of WT bacteria-mediated tumor suppression, e.g. Figure 3e shows that even WT bacteria have a significant suppressive effect on tumor growth. Could the authors provide background on what is known about the mechanism of this tumor suppression, outside of tumor targeting and engineerability? They only reference "immune system stimulation."

We appreciate the reviewer’s comment and would like to refer the reviewer to our recently published article [Lim et al., EMBO Molecular Medicine 2024; DOI: 10.1038/s44321-023-00022-w], which shows that in addition to immune system stimulation, WT bacteria can also be perceived as an invading species in the tumor that can exert differential selective pressure against cancer cells. Competition for nutrient is highlighted as a major contribution to contain tumor growth. In fact, the nutrient competition that we observed in the prior article inspired the design of the iron scavenging bacteria towards overcoming nutritional immunity. We have cited this recently published article to the revised manuscript to enrich the background.

The authors claim that there is immunologic memory because of tumor resistance in re-challenged mice after IroA-*E. coli* treatment (Fig 5c). It appears that the control group for this experiment is naive mice (and not WT-*E. coli* treated mice), in which case the immunologic memory could be from having had tumor*/E. coli* rather than the effect of IroA-*E. coli*.

We have modified our claims to highlight that the tumor eradication by iron scavenging bacteria can establish adaptive anticancer immunity through the elicitation of CD8 T cells. We did not intend to highlight that the adaptive immunity stemmed from IroA-*E. coli* only, and we intend to build upon current literature that has reported CD8+ T cell elicitation by bacterial therapy. The IroA-E.coli is shown to enhance adaptive immunity. We also did not consider a WT-*E. coli* control as no WT-E. coli treated group achieved complete tumor regression.

The authors claim that CD8+ T cells are mechanistically important in the effects of iron status manipulation in *E. coli*-mediated tumor suppression (Fig 5). In order to show this, it seems that Fig 5c should include WT-*E. coli* and WT-*E. coli*+CD8 ab groups, as it may be that CD8+ T cells play a role in tumor suppression regardless of whether or not iron regulation is being manipulated.

We apologize for the confusion from our prior writing. We have modified our claims to highlight that the tumor eradication by iron scavenging bacteria can establish adaptive anticancer immunity through the elicitation of CD8 T cells. We did not intend to convey that CD8+ T cells are mechanistically important in the effects of iron status manipulation.